# Metabolites of Seaweeds as Potential Agents for the Prevention and Therapy of Influenza Infection

**DOI:** 10.3390/md17060373

**Published:** 2019-06-22

**Authors:** Natalia Besednova, Tatiana Zaporozhets, Tatiana Kuznetsova, Ilona Makarenkova, Lydmila Fedyanina, Sergey Kryzhanovsky, Olesya Malyarenko, Svetlana Ermakova

**Affiliations:** 1Federal State Budgetary Scientific Institution, Somov Research Institute of Epidemiology and Microbiology, Sel’skaya street, 1, Vladivostok 690087, Russia; besednoff_lev@mail.ru (N.B.); niiem_vl@mail.ru (T.Z.); takuznets@mail.ru (T.K.); ilona_m@mail.ru (I.M.); 2Far Eastern Federal University, School of Biomedicine, bldg. M25 FEFU Campus, Ajax Bay, Russky Isl., Vladivostok 690922, Russia; fedyanina.ln@dvfu.ru (L.F.); kryzhanovskii.sp@dvfu.ru (S.K.); 3G.B. Elyakov Pacific Institute of Bioorganic Chemistry, Far Eastern Branch of the Russian Academy of Sciences, Pr. 100-letiya Vladivostoka, 159, Vladivostok 690022, Russia; malyarenko.os@gmail.com

**Keywords:** Flu, alga, sulphated polysaccharides, alginates, lectins, polyphenols, anti-viral activity

## Abstract

Context: Seaweed metabolites (fucoidans, carrageenans, ulvans, lectins, and polyphenols) are biologically active compounds that target proteins or genes of the influenza virus and host components that are necessary for replication and reproduction of the virus. Objective: This review gathers the information available in the literature regarding to the useful properties of seaweeds metabolites as potential agents for the prevention and therapy of influenza infection. Materials and methods: The sources of scientific literature were found in various electronic databases (i.e., PubMed, Web of Science, and ScienceDirect) and library search. The retrospective search depth is 25 years. Results: Influenza is a serious medical and social problem for humanity. Recently developed drugs are quite effective against currently circulating influenza virus strains, but their use can lead to the selection of resistant viral strains. In this regard, new therapeutic approaches and drugs with a broad spectrum of activity are needed. Metabolites of seaweeds fulfill these requirements. This review presents the results of in vitro and in vivo experimental and clinical studies about the effectiveness of these compounds in combating influenza infection and explains the necessity of their use as a potential basis for the creation of new drugs with a broad spectrum of activity.

## 1. Introduction

The socioeconomic losses associated with viral infections of the respiratory tract are enormous at present. In Russia, acute respiratory infections (ARIs), including influenza, occupy a leading position for infectious diseases [1]. According to the Russian Federal Service for Surveillance on Consumer Rights Protection and Human Wellbeing (Rospotrebnadzor), 27.3–41.2 million cases of these diseases are registered in Russia each year. The total economic damage from ARIs in Russia ranges from 40 to 100 billion rubles annually [2]. In the United States, more than 400,000 people per year are hospitalized with respiratory viral infections [3].

Influenza is a serious medical and social problem for humanity. This acute infectious disease is caused by an enveloped ribonucleic acid (RNA) containing virus belonging to the family Orthomyxoviridae. Every year, more than 500 million people in the world get the flu; about 2 million of them die [4]. In Russia, every seventh person is involved in the influenza virus’ annual epidemic [5]. When a new antigenic variant of the virus appears, a new pandemic covers all regions of the Earth. It is characterized by high morbidity with a large number of patients requiring hospitalization and mortality among all age groups of the population [6]. The 2009 pandemic of influenza A (H1N1)pdm09 was accompanied by wide coverage of the population of many countries of the world, including the Russian Federation, with serious clinical complications and high mortality [7,8]. Adverse outcomes were observed not only among immunocompromised individuals in high risk groups but also in healthy young people with no significant previous pathology, including pregnant women. The influenza virus can cause disease not only in humans but also in various animals (e.g., birds, pigs, horses) [9]. In recent years, cases of human infection with avian influenza viruses of the H5N1, H7N7, and H7N9 subtypes have been reported [10].

Despite the presence of a significant number of anti-influenza drugs, this infection is still dangerous, as annual flu epidemics remain insufficiently controlled. Recently developed drugs are quite effective against currently circulating influenza virus strains, but their use can lead to the selection of resistant viral strains [11]. In this regard, there is a need for new therapeutic approaches and drugs with a broad spectrum of activity. Sulphated polysaccharides of algae (red and brown), such as carrageenans and fucoidans fulfill these requirements. This review presents the in vitro and in vivo results of experimental and clinical studies demonstrating the excellent effectiveness of these compounds in combating influenza infection and explain the need to use them as a potential basis for the creation of new drugs with a broad spectrum of activity. Despite their pronounced antiviral properties, they have not been entered the category of drugs yet, because of the difficulties of these compounds’ standardization [12].

The purpose of this review is to draw the attention of researchers working on the problem of prevention and therapy of influenza to coordinate their efforts for the creation of standard samples of these highly active biopolymers.

## 2. Influenza Virion

Influenza virions are particles with a diameter of 80–100 nm, coated with a lipid membrane with an integrated surface of three types of glycoproteins: hemagglutinin (HA), neuraminidase (NA), and the viral ion channel (M2) [13]. On one side, this membrane is in contact with the cytoplasmic domains of HA and NA, and on another, with the core of the virion. The ribonucleoprotein (RNP) is represented by eight segments of the genome: single-stranded negative-polarity RNA in a complex with the nucleoprotein protein and three subunits of the polymerase complex [14,15]. After the virus enters the cell, the ribonucleoprotein gets to the nucleus, where transcription, translation, and replication of segments of the viral genome occur (Figure 1).

The binding of the influenza virus with cell sial-containing receptors creates surface glycoproteins, i.e., hemagglutinin (HA), neuraminidase (NA), and viral ion channel (M2). Influenza virus penetration into the cell and fusion with endosome: HA contributes to the release of free nucleocapsid in the cytoplasm (the so-called fusion sites). M2 is involved process of "disassembly" of the virus in endosomes and the Golgi apparatus. Transcription and replication of the viral genome creates transcription and replication of virus-specific RNAs carried out by a viral polymerase complex. Ribonucleoprotein (RNP) enters the cytoplasm and then into the nucleus. The release of the RNA virus in the cytoplasm allows for NA to end the replication and helps to separate matured virion from epithelial cells. Further, the matrix protein (M1) comes into contact with the cytoplasmic domains of HA, NA, and the core of virion. M1 plays an important role in the assembly and disassembly processes of new virus particle.

Transcription and replication of virus-specific RNAs are carried out by a viral polymerase complex. Unlike the polymerases of eukaryotic cells, viral polymerase has no error correction mechanism. Therefore, the frequency of mutations of the viral genome is, according to various estimates, from 10^−4^ to 10^−6^ nucleotides per replication cycle [16,17]. This is several orders of magnitude higher than the rate of mutation in bacteria and eukaryotes [18]. Due to a short life cycle, the evolution of the influenza virus is fast. As a result, the rapid appearance of mutations allows the virus to escape the host’s immune response [19,20]. Thus, it can cause annual epidemics, despite the formation of an immune layer in the population due to vaccination and natural incidence. In addition, as result of the use of antiviral drugs, drug-resistant strains of the virus have developed, leading in a decrease in the effectiveness of antiviral chemotherapy [21].

Influenza virus, like all complex viruses, has a supercapsid, i.e., an additional envelope or peplos, the structural elements of which are peplomer, including HA and NA [22]. The capsid encloses the genome of the virus. Three types of influenza virus (genera) have been described: A, B, and C [23]. Type B and C viruses cause disease only in humans. Hemagglutinin and neuraminidase carry antigenic determinants of the influenza virus and determine its subtype (H5N1, H3N2, H1N1, etc.). Influenza hemagglutinin is a highly variable surface glycoprotein, with 17 known antigenic subtypes [24]. The main function of hemagglutinin is receptory, i.e., it mediates the attachment of virions to target cells by binding to sial-containing receptors [25]. Hemagglutinin is the main specific antigen of the virus, causing the formation of antibodies that neutralize the infectivity of the virus. The presence of antibodies against HA is the main criterion for assessing the post-infectious or post-vaccination humoral immune response to the influenza virus [21].

However, the value of HA is not limited to physical contacts between the virus and the cell. It continues to act in the subsequent stages of infection, contributing to the release of free nucleocapsid into the cytoplasm. This occurs in the acidic environment of endosomes (phagolysosomes), due to the special structures of HA (its so-called fusion sites) that promote the unification of viral and cellular membranes.

The NA of influenza destroys sialic acid-based cell receptors on infected cell surfaces and on virions after generation, and thereby prevents virion self-aggregation, facilitating the passage of the virus through mucin during a natural infection. In influenza A viruses, there are nine subtypes of NA, but there is only one NA in B and C viruses. The NA separates the virions from sialylated mucins that cover the mucosa, promoting virus attachment to the surface of epithelial cells. At the end of the replication cycle, it helps to separate mature virions from epithelial cells. In both cases, NA acts as a spreading factor, expanding the area of infection. That is why antibodies to NA show a protective effect, but weaker than antibodies against HA [26].

The nucleoprotein of influenza virus (NP) is the main antigen recognized by cytotoxic T lymphocytes. Amino acid sequences 260–283 of the nucleoprotein of influenza A virus induce the T cell response. The NP of influenza virus is a major factor in the cycle of viral infection when switching the synthesis of influenza virus RNA from transcription mode to replication mode [25].

Protein M1 is the matrix protein of the influenza virus; it covers the lipid membrane. It is in contact with the cytoplasmic tails of HA and NA, and it is bound to the ribonucleoprotein complex of the virus [27]. This protein plays an important role in the assembly and disassembly processes of new virus particles [28]. Membrane protein M2 is an influenza virus surface protein that participates in the creation of an ion channel that regulates pH while disassembling the virus in endosomes and the Golgi apparatus. An acidic pH is a prerequisite for disassembly the virus and stabilizing it during intracellular transport. Viruses with a defect in the activity of this ion channel have poor reproduction efficiency [29].

## 3. Pathogenic Targets of Influenza Viruses in Humans

Cells of the single-layer multi-row cylindrical ciliated epithelium of the respiratory tract are main targets of the influenza virus. For infection, the virus must overcome the factors of non-specific resistance of the organism such as the viscous properties of mucus, the movement of the cilia of the cylindrical epithelium, action of non-specific inhibitors of viral replication in mucus, the macrophagal barrier and IgA [30]. With the help of HA, the virus attaches to the receptors of target cells and penetrates them, where the replication cycle takes place [5]. After 4–6 h, a batch of new viruses appears in the cell and then they are pushed out through the cell membrane to the outside. After 24 h, the number of viruses whose “parent” entered the cell can reach several hundred million. The released virions infect neighboring cells and some of them enter in the blood. The subsequent death of epithelial cells is caused not only by the cytopathogenic effect of the virus but also by the inability of the cell to fully recover after the replication of the pathogen. In the early stages of infection, Toll-like receptors (TLR) are involved in the recognition of conservative molecules characteristic of pathogenic microorganisms. The influenza virus stimulates the TLR responses of innate and adaptive immunity [31]. On the cell membrane, TLR4 and C-type lectin receptors (MMR-mannose and MGL-galactose) interact with the glycoproteins of the viral membrane, HA, and NA [32]. Inside cells, the endosomal receptors TLR3, TLR7, TLR10, and cytoplasmic RIG1 respond to viral RNA and RNP [33,34].

Since this article summarizes obtaining experimental data on potential next generation of anti-viral agents: biopolymers from brown, red, and green algae, the above-discussed information concerning virus action is important for understanding of mechanisms of the action of marine sulphated polysaccharides on the influenza virus and its targets in the body.

## 4. Recent Anti-Influenza Drugs

Despite the use of a significant number of anti-influenza drugs in medicine of different countries, the search for and development of new, effective and whenever possible harmless drugs continue [35,36]. Currently, for the prevention and treatment of influenza infection, a number of drugs are available with different mechanisms of action, such as limiting the infection at the early stages, stimulating innate immunity, and manipulating the effect of interferons (Table 1).

For a long time, rimantadine and amantadine were very actively used to treat the flu. Leibbrand et al. [37] reported that these anti-influenza drugs act on two stages of the life cycle of the influenza virus: stripping the virus and releasing virus particles from the cell after replication. Rimantadine is able to increase the acidity inside vacuoles surrounding viral particles after they penetrate into the cell. As a result, the fusion of the influenza virus with the vacuole membrane is prevented, which makes it impossible to transfer viral genetic material from the virus particle to the cytoplasm of the cell. Thus, rimantadine becomes an obstacle between the influenza virus and the genome of the host. Along with clinical efficacy, these drugs caused side effects in the gastrointestinal tract and the nervous system [38]. In addition, many influenza A viruses (for example, H3N2, H1N1) quickly became resistant to these agents [39].

At present, adamantane group of drugs are recommended in case all other measures fail. In the last decade, a number of drugs with direct action on the influenza virus (direct-acting antivirals, DAA) have appeared [39]. These include NA inhibitors (oseltamivir/Tamiflu^TM^ and undecivir/Relenza^TM^), M2 channel blockers, and fusion inhibitors (umifenovir/Arbidol^TM^). Zanamivir and oseltamivir have shown lower than expected efficacy in clinical trials. At the same time, these drugs are more effective in children, in whom the drugs reduce the duration of the disease if they are applied within 48 h after the onset of the first symptoms; they also reduce the number of serious complications. Moreover, these drugs are successfully used to prevent and reduce the intensity of symptoms of seasonal flu. The disadvantages of these chemotherapy drugs are a decrease in their effectiveness during the later stages of infection [40]. In cases of prolonged, complicated flu, pathogenic therapy preparations are used to eliminate the effects of the cytokine storm, reducing the severity of reactive processes [15]. Such processes are induced by the virus, but are carried out by the host mechanisms. These include toxic shock syndrome, hyperproduction of proinflammatory cytokines, cellular tissue infiltration, haemorrhagic syndrome, etc. Although modern drugs are quite effective against currently circulating influenza virus strains, their use often leads to the selection of resistant viruses and resistant strains [11].

In this regard, we need new therapeutic approaches and broad-spectrum drugs. Drugs aimed at various viruses or impairing different stages of influenza virus replication will represent a serious means of fighting infections and minimizing the development of resistant viruses. The sulphated polysaccharides of algae (red, brown and green), such as carrageenans, fucoidans, and ulvans, have been studied as anti-influenza drugs and met these requirements. At the same time, there are other compounds from algae that also have antiviral effects, including anti-influenza action, which should be subjected to further experimental and clinical study.

## 5. Polysaccharides

These compounds represent a class of biopolymers, the content and structure of which vary depending on the type of algae, its place of growth, climatic conditions, harvest season, method of extraction and many other factors [41]. According to the literature, sulphated polysaccharides (fucoidans, galactofucans, dextran sulphates, carrageenans, sulphated chitosans, synthetic polyvinyl, and polyethylene sulphates) have antiviral activity not only against the influenza virus but also against many other viruses, such as those that cause hepatitis C, tick-borne encephalitis, haemorrhagic fever with renal syndrome, dengue fever, and AIDS. It is known that, in the human body, the most prevalent heteropolysaccharides are glycosaminoglycans, negatively charged long unbranched polysaccharides consisting of disaccharides repeating units [42]. The binding of glycosamines with different ligands leads to post-translational modifications that facilitate cell migration, proliferation, and differentiation. The glycosaminoglycans, a class of heparan/heparan sulphates present in the basement membrane, in the extracellular matrix, and on the cell surface that are able to specifically interact with macromolecules of the extracellular matrix (fibronectin and laminin), enzymes, and an extensive class of heparan-binding molecules (growth factors and chemokines) [42]. Mimetics of glycosaminoglycan, including heparan/heparan sulphates provide a wide range of biological effects and modulate the effect of many signaling molecules in cells [43]. Sulphated polysaccharides of algae are natural mimics of heparan sulphates. Fucoidans and carrageenans can mimic the action of endogenous factors and regulate the functions of microorganism systems through key cell and enzyme receptors. Due to this, sulphated polysaccharides have the ability to bind to various receptors on the surface of the host cell and compete with viruses for glycoprotein receptors [44].

One of the characteristic features of algal polysaccharides is the presence of sulphate groups and uronic acid residues in their structures, which distinguishes them from the polysaccharides of land plants [45]. In the last decade, quite a lot of scientific articles have appeared that presents the effectiveness of fucoidans and carrageenans in influenza infection, as well as acute viral “colds”.

### 5.1. Carrageenans

Carrageenans are sulphated polysaccharides of red algae whose chemical structure is based on a disaccharide repeating unit consisting of two D-galactose residues joined with each other by β-1,4 glycosidic bonds. These units are bound in polysaccharides by α-1,3 bonds [46]. The structural diversity of carrageenans is due to the presence of β-(l,4)-linked residues in the form of 3,6-anhydrogalactose, as well as the number and position of sulphate groups in monosaccharide residues [47]. Regular polysaccharides, the polymer chain of which is built from repeating disaccharide units of the same type, represent a different class of carrageenans. Natural carrageenans are rarely regular, as more often they contain repeating units of several types and have an irregular or hybrid structure (Figure 1), structures of which are explained by the multi-stage biosynthesis of polysaccharides in the algal cell wall.

The variability of the primary structure of carrageenans determines the diversity of their macromolecular organisation and defines a wide range of their biological activity [46]. The uniqueness of these hydrocolloids is in the alternating galactose and 3,6-anhydrogalactose residues, which are linked by α-1,3 and β-1,4 glycosidic bonds. A characteristic feature of carrageenan molecules is the large number of sulphate groups [48].

The quantity and location of sulphuric acid residues determine the type, form, and functions of carrageenans, which are actively used in food industry for the production of meat, dairy, and confectionery to improve the microtexture of nutritive products, i.e., as gelling agents, emulsifiers, and thickeners. Among the polysaccharides of algae, carrageenans are the most studied regarding their toxicity, pyrogenicity, and allergenicity [49]. The safety of their use in food and medical purposes has been confirmed by numerous studies [50]. Among the diverse biological properties of these sulphated polysaccharides, their antiviral, anticoagulant, immunomodulatory, antitumor, and anti-ulcer activity are currently attracting the greatest interest [46,51,52]. Sulphated polysaccharides interact with a variety of eukaryotic cell proteins and have a multidirectional effect on the body’s immune response, both inhibitory and stimulatory, which makes it possible to consider carrageenans as possible immunomodulators. It is assumed that the immunomodulatory effect of carrageenans is initiated by α-Gal-(1,3)-Gal epitopes [53]. Recently, new data have appeared on the antioxidant activity of algae polysaccharides [46,54].

Carrageenans have attracted the attention of researchers investigating the problem of influenza and other acute respiratory viral infections, particularly in terms of the possibility of creating a physical barrier in the nasal cavity against respiratory viruses, including the flu virus [55]. For this purpose, kappa (k), iota (ι), and lambda (λ) carrageenans were used. It is known that carrageenans block the interaction of viruses with cells and also inhibit the formation of a syncytium, induced by influenza A viruses. A. Leibbrand et al. [37] carried out an investigation showing the effectiveness of carrageenan against human influenza A viruses. The authors determined the sensitivity of the H1N1 influenza virus strains, as well as the pandemic H3N2 strain, to carrageenan subtypes ι and k using the plaque formation method in canine kidney epithelial cells (MDCK). The most active in this test was ι-carrageenan (IC_50_ or 50% inhibitory concentration = 0.04 μg/mL); k-carrageenan was less active (IC_50_ = 0.3 μg/mL). The purity of the ι- and k-carrageenans used in these studies was above 95%, and the molecular weight of both polymers was more than 100,000 Da. At the concentrations of 40 and 4 µg/mL, ι-carrageenan effectively reduced viral replication by 2–4 log units within 96 h after infection. Thus, it was found that ι-carrageenan contributes to the survival of cells infected with the virus by direct exposure to the virus. In another series of experiments, the same authors investigated the effect of carrageenans on an influenza virus-infected primary cell culture of the human epithelium from the nasal cavity. Under these conditions, ι-carrageenan inhibited the formation of plaques by the pandemic strain H1N1/2009 (IC_50_ about 0.04 µg/mL). At the same time, an interesting fact was established: to obtain an effect when cells were infected with another virus (A/PR8/34 H1N1), a higher concentration of polysaccharide was required, i.e., the sensitivity of different strains to carrageenan was different.

Carrageenans are high molecular weight compounds and therefore it is unlikely that they can pass through the barriers of the body. However, local administration has a pronounced effect, for example, with influenza infection and other viral diseases of the respiratory system. In this case, carrageenans reduce the spread of the virus in the surface epithelium of the respiratory organs of infected animals and contribute to survival.

Unfortunately, the solubility of carrageenans is limited, especially in aqueous solutions containing potassium and calcium ions [52], since in their presence carrageenans form viscous gels. Another disadvantage of carrageenans is their anticoagulant properties. Despite this, ι-carrageenan has passed clinical trials and a nasal spray based on it has already been successfully sold in Europe for use in viral infections of the respiratory tract in humans. The effectiveness of the spray for ARIs was also reported by Eccles et al. [56]. The authors showed that, compared with persons receiving placebo, patients in the experimental group noted such significantly reduced symptoms of the disease as nasal congestion, runny nose, cough, and sneezing. Moreover, nasal congestion at the end of the observation period was noted by 63.6% of persons in the placebo group and 28.6% of the group receiving carrageenan. Viral capacity in the nasal mucosa in patients treated with the spray was significantly decreased (92%), while placebo treatment did not affect viral replication. The nasal spray was effective when used during the first 48 h after the onset of symptoms. Similar results were obtained by Ludvig et al. [57].

Spray application reduced the expression of pro-inflammatory cytokines and increased the level of IL-1 and IL-12p40 receptor antagonists, which are known to have anti-inflammatory action in the nasal lavage of patients with respiratory viral infections [56]. It is known that IL-12p40 is necessary for inhibiting the hyperactivity of airway and peribronchial fibrosis [58]. The expression of inflammatory mediators during viral infection may complicate underlying diseases in the form of asthma [59,60]. In this regard, a decrease in the intensity of the immune response due to a lower viral load seems to be an attractive property of treatment of ι-carrageenan. To increase their efficiency, oligosaccharides and their sulphated derivatives having a lower molecular weight were obtained from the high molecular weight ATP carrageenan [61]. So, the oligosaccharide CO-1 with a molecular weight of 1–3 kDa effectively dose-dependently inhibited the replication of influenza A (H1N1) virus in MDCK cells (selectivity index >25.0). CO-1 did not bind to the cell surface, but it was bound to viral particles during the pre-treatment process. Unlike high molecular weight native carrageenan, this oligosaccharide can penetrate into MDCK cells and inhibit the expression of viral proteins and mRNA after its internalization into the cell, but before it leaves the cell, i.e. in one replication cycle. The main factors affecting the antiviral activity of oligosaccharides are the degree of sulphation and Mw. The most active oligosaccharide CO-1 contained 0.8–1.0 mol/mol sulphate, and its molecular weight was 1–3 kDa. The preparation CO-1 and its full sulphated derivative (COS) significantly increased the survival rate of mice infected with a lethal dose of influenza virus and reduced the viral load in the lungs of these animals [62]. Taking into account these findings, the authors proposed using low molecular weight oligosaccharides of carrageenan in the treatment of influenza as an alternative strategy to combat this infection.

Shao et al. [63] investigated the molecular mechanisms of cell protection using k-carrageenan against SW731 influenza virus penetration. The authors showed that the polysaccharide specifically and effectively inhibited the reproduction of the influenza virus. The MDCK cells were infected with various strains of the influenza virus, after which they were treated with carrageenan at different doses. After 24 h, a dose-dependent decrease in the titer of SW731 and CA04 viruses (homologous MDCK) was recorded. The remaining experimental strains of influenza virus (PR8, WSN, ZB07, and H1N1) were insensitive to carrageenan. Thus, k-carrageenan prevented the development of the extra- and intracellular stages of influenza virus replication. To determine the stage of reproduction of the virus affected by carrageenan, the authors added polysaccharide to infected cells during the period of adsorption (0 h), internalization (1 h), early replication (2–6 h), and release (8 h). Then, after 24 h, the inhibitory effect was evaluated, in response to treatment at the time points of 0, 1, 2, 4, and 6 h. After 8 h, there was no inhibition. Thus, it was shown that not only extracellular, but also some intracellular stages of influenza virus replication were affected by k-carrageenan. The titer of influenza virus SW731 was decreased in cases where the virus was treated with the polysaccharide before or during infection of the cells. When treating cells with carrageenan, there was no such effect, i.e. the best results were obtained by the action of the polysaccharide at the adsorption stage. The same authors showed that carrageenan does not inactivate the influenza virus, since no differences were found between the pre-treatment group and the adsorption group. Carrageenan specifically inhibited the binding of the HA virus to its receptor, sialic acid. The effect of polysaccharides on the exit stage was insignificant. The authors have proposed the use of carrageenan against the H1N1/2009 influenza virus and other viruses containing HA/H1N1/2009. The carrageenan used in these experiments contained both high- and low-molecular components, and therefore it acted on the extracellular and intracellular stages of reproduction.

Other authors [64], investigating the polysaccharides of the red alga *Gyrodinium impudicum*, obtained sulphated galactan conjugated with uronic acid and studied its activity as an anti-influenza agent. As in the studies by Wang W. [61,62] it was shown that the antiviral activity (IC_50_ against the influenza virus at doses 0.19–0.48 µg/mL) of the galactan is related to its ability to interact with viral particles, which preventing virus adsorption and internalization.

Yu et al. [65] suggested using hybrid carrageenan (ı/κ/ν-carrageenan) as a potential inhibitor of influenza A virus. In this study, the authors obtained three polysaccharides from the red alga *Eucheuma denticulatum* by successive extraction with cold and hot water and an aqueous solution of NaOH: hybrid polysaccharide (EW), preparation EH, containing only ι-carrageenan, and α-1,4α-d-glucan (EA), which consisted of 88% of glucan and 12% carrageenan as an impurity. The molecular weights of the compounds were 480, 580, and 510 kDa, respectively. Antiviral activity against the H1N1 influenza virus was highest when used the hybrid polysaccharide (276.5 μg/mL), and the H1N1 virus suppression index was 52% using a polysaccharide dose of 250 μg/mL. The IC_50_ for ι-carrageenan EH was 366.4 μg/mL. The polysaccharide EA showed the lowest antiviral activity (IC_50_ > 430 µg/mL).

The study of Fazekas T. et al. [66] was very important, because it was conducted in a clinical setting with the participation of patients (children and adults aged from 1 to 18 years) with respiratory viral infections, including influenza B. Intranasal spray was used three times a day in for seven days. Symptom dynamics was monitored and viral load was determined. In this study iota-carrageenan did not alleviate symptoms in children with acute symptoms of common cold, but significantly reduced viral load in nasal secretions that may have important implications for future studies. In this study, v-carrageenan as part of the spray did not reduce the severity of symptoms in children with acute cold symptoms, but significantly reduced the viral load in the nasal lavage of patients who received the spray compared with the control group (27% versus 13%, respectively).

A positive evaluation of the effectiveness of nasal sprays in patients with acute respiratory viral infections was provided in two randomized double-blind, placebo-controlled trials by Koenighofer et al. [67]. In patients treated with carrageenan, the duration of the disease decreased by two days, there were fewer relapses, and the body was cleared of viruses more rapidly. The spray was effective in both children and adults. The treatment of patients with influenza by carrageenan gel was shown to significantly facilitate respiratory tract, reduces the duration of illness, and the severity of symptoms of intoxication. Carrageenans were found to provide a more pronounced synergistic effect with anti-influenza drugs with a different mechanism of action.

A number of authors have proposed increasing the effectiveness of the treatment by combining carrageenan with other drugs. Thus, a combined intranasal spray, including carrageenan and zanamivir (an NA inhibitor) was proposed by Morokutti-Kurz et al. [68]. Previously, the authors investigated the efficacy of in vivo and in vitro intranasal administration of zanamivir in different doses for the prevention and treatment of influenza. Their study showed that treatment of animals before infection and 36 h after infection with a virus was not accompanied by adverse events.

Zanamivir and carrageenan separately exhibited different antiviral activity against different strains of the influenza virus. Since the mechanism of action of these agents is quite different, one could expect protection against a wider spectrum of viruses than with their individual use. Both compounds and the complex preparation were non-toxic at the highest concentration (400 μg/mL zanamivir and 533 μg/mL carrageenan). The effectiveness of the suppression of the replication by both substances depended on the virus strain. The IC_50_ value for zanamivir ranged from 0.18 µg/mL for H5N1 and 22.97 µg/mL for H7N7. The IC_50_ values for carrageenan ranged from 0.39 µg/mL to 118.40 µg/mL for H1N1 and H7N7, respectively. Thus, zanamivir and carrageenan target different strains of influenza virus to varying degrees and, therefore, they can provide broader anti-influenza activity by acting synergistically. At the same time, the physical interaction of carrageenan with the virus did not violate the inhibition of NA by zanamivir. The effectiveness of the spray increased when ı-and k-carrageenans were used simultaneously. Mice infected with a lethal dose of the influenza virus that received the placebo dies, as did the animals in all groups receiving monotherapy; however, the combined spray statistically significantly increased the survival of animals. The authors believed that if a vaccine does not keep up with a virus that has changed its composition, such a spray will to some extent protect the population from the impending epidemic.

In addition, an interesting combination of two drugs was suggested [69]. For more than 50 years, xylometazoline has been used to relieve vasoconstriction and oedema of the nasal mucosa in acute respiratory viral infections caused by a wide variety of respiratory viruses, including the influenza virus. The authors combined this vasoconstrictor and ı-carrageenan in one preparation, which had an antiviral effect. It was found that the polysaccharide did not reduce the efficacy and safety of xylometazoline, and the antiviral efficacy of ı-carrageenan remained unchanged.

Thus, carrageenan is currently widely used as a therapeutic and prophylactic agent and was also proposed as an integral part of various antiviral drugs. On the other hand, there is evidence in the literature that oral administration of carrageenan by laboratory animals can lead to the development of inflammation of the gastrointestinal tract [52,70]. However, carrageenan compounds are considered safe and approved for use [48,71]. The Joint Expert Committee on Food Additives (JECFA) concluded that the use of carrageenan is acceptable, even in childhood: “The use of carrageenan in a formula for children or for special medical purposes at concentrations up to 1000 mg/mL does not cause concern” [72]. This confusion, it seems, may be due to imperfect terminology. Some authors combine low molecular weight products of carrageenan hydrolysis, such as polyginan and degraded carrageenan, which are clearly toxic, and native non-degraded food carrageenan, which is considered safe, under the general term “carrageenan” [71]. However there have already been reports of the ability of dietary λ-carrageenan to cause enteritis in rats with prolonged oral ingestion [70].

Among the many study associated with influenza infection, we did not find reports of adverse side effects of carrageenans at intranasal use. Moreover, in Europe, as already mentioned above, the use of the spray for intranasal administration as a preventive and therapeutic agent for influenza and ARIs is permitted. Apparently, attention should be paid to reports of negative phenomena associated with the use of carrageenans and to study this issue separately with respect to these infections.

Thus, it was reported that carrageenans created a physical barrier in the nasal cavity against respiratory viruses, including various strains of influenza virus. At an early stage of viral infection carrageenans are directly associated with the influenza virus, preventing its adsorption, penetration, and replication. At the same time, the antiviral effect of carrageenans is specific and is due to the screening of the cellular structures involved in the binding of the virus to its receptors.

### 5.2. Fucoidans

Fucoidans are highly sulphated, usually branched polysaccharides, often containing, in addition to fucose residues, glucose, galactose, xylose, mannose, and uronic acids, as well as acetyl groups [45]. The structures of polysaccharides from brown algae are very diverse and depends on the type of alga, its reproductive status, and other abiotic factors. In fact, each new polysaccharide isolated from algae is a new substance, and its molecule contain unique structural elements. This is why the determination of the structure of fucoidans, as well as the clarification of the structure/function relationship of these polysaccharides, is extremely difficult. 

It should be noted that the effectiveness of fucoidans of brown algae as potential anti-influenza agents has been studied quite actively in recent years due to the polyvalence of their effects (antiviral, antibacterial, anti-inflammatory, immunomodulatory effects, etc.), as well as the fact that fucoidans penetrate biological membranes. Efficiency of oral administration of fucoidan is confirmed by data of its transformation in macroorganism, which are presented in articles [73,74]. The possibility of the appearance of fucoidan derivatives in the peripheral blood was confirmed by Irhimeh M.R. et al. [73]. Authors using monoclonal antibodies to highly sulfated fucoidan found its derivatives in plasma of healthy participants who took orally for 12 days at 3 g/d Undaria algae powder, containing 10% of fucoidan derivatives and purified galactofucane sulfate. The average concentration of fucoidan detected in plasma was 4.002 mg/L and 12.989 mg/L, respectively. Tokita Y. et al. [74] also found fucoidan from Cladosiphon okamuranus in the serum and urine of healthy participants 6 and 9 h after ingestion of the polysaccharide orally. These facts indicate the possibility of degradation of fucoidan molecules in the human body and the participation of its derived structures in the implementation of antiviral properties.

After administration, these polysaccharides can be detected in the urine and serum [74,75]. Histological studies using monoclonal antibodies against fucoidan made it possible to detect it in the small intestine, the epithelial cells of the jejunum, in mononuclear cells, and in sinusoidal non-parenchymal cells of the liver [76]. The same authors established the active transport of fucoidan through a monolayer of Caco2 cells in vitro and the excretion of fucoidan in the urine of a patient after oral administration. The level of fucoidan increased from 3 to 9 h after administration [77]. To prevent the destruction of fucoidan in the stomach, it is suggested to enclose it in chitosan nanocapsules.

A study on the anti-influenza activity of polysaccharides from the sporophylls of the brown alga *Undaria pinnatifida* allowed Synytsya et al. [78] to establish that in mice, infected in vivo with avian influenza A viruses (subtypes H5N3 and H7N2), the level of virus replication decreased and the production of specific antibodies increased. Oral administration of the polysaccharide blocked the release of the virus from cells and significantly increased the titer of virus-neutralizing antibodies and IgA. This polysaccharide presents as a low molecular weight (Mw 9 kDa) fucogalactan, consisting of partially sulphated and acetylated fucose and galactose residues in approximately equal amounts and having a complex structure. Previously, Hayashi et al. [79] investigated the effectiveness of this *O*-acetylated sulphated fucogalactan in immunocompetent and immunocompromised mice infected with a lethal dose of influenza virus. The use of this polysaccharides reduced virus replication, weight loss, and mortality in animals of both groups and increased their lifespan. Oral administration of fucoidan caused an increase in the titre of neutralizing antibodies in the blood and mucous membranes. In immunocompromised mice, drug-resistant viruses often multiply after treatment with oseltamivir. No resistant viruses were isolated from mice treated with fucoidan. The authors proposed the combined treatment with oseltamivir and fucoidan, because in this case there was no recurrence of influenza virus reproduction, as is sometimes the case when treating only with oseltamivir. Combined treatment with fucoidan and oseltamivir was thus recommended by the authors as a new treatment strategy for influenza infection.

The fucoidan from the brown alga *Kjelmaniella crassifolia* (Mw about 536 kDa, sulphate content 30.1%, purity more than 98%) is a glucuronomanan with branches in the form of oligosaccharides at position 3 of the fucose residues. Oligosaccharides (degree of polymerisation from 0 to 6) consist of 3-linked glucose residues, sulphated at positions 2 and 4 [62]. Intranasal (for four days) application of fucoidan increased the survival of mice (80% versus 30%) and their lifespan and reduced the viral load of the lungs in influenza-infected animals compared to the control group (*p* < 0.05). When treated with oseltamivir alone, 90% of the mice survived. All influenza viruses used in the experiment were sensitive to treatment with fucoidan, but the most susceptible virus was H1N1 (Ca109) (IC_50_ < 6.5 μg/mL). Treatment with fucoidan reduced the severity of flu symptoms and pathological changes in the lungs. One valuable quality of fucoidan was the lack of formation of resistant strains of the virus under the action of this polysaccharide. In the supernatants of spleen cells, the levels of interferon-gamma (IFN-γ) and interleukin-2 (IL-2) increased following treatment with fucoidan compared with the control animals. In addition, a direct effect of fucoidan on viral particles was found. It was shown that pre-incubation of the virus with fucoidan at concentrations of 31.25–250 μg/mL significantly reduced the number of plaques in MCDK cell culture, i.e. this polysaccharide can inactivate viral particles by direct contact. This polysaccharide inhibited the activation of the epidermal growth factor receptor (EGFR-epidermal growth factor receptor) and was able to bind to viral NA and inhibit its activity. In this regard, the authors believed that such inhibitors of the EGFR pathway and NA can be used alone or with other drugs to block the processes of penetration and the release of influenza A virus from cells. The investigated fucoidan is a potential candidate for creating a medicine in the form of a spray or drops.

Using SPEV cell culture sensitive to the reproduction of influenza A (H5N1), Makarenkova et al. [80] investigated the in vitro antiviral effect of a fucoidan from the brown seaweed *Laminaria japonica* against the H5N1 influenza virus. The results showed that the fucoidan did not possess cytotoxic properties in concentrations from 500 μg/50 μL to 125 μg/50 μL and did not change the morphological properties of the SPEV cell culture. Fucoidan had a virucidal effect and suppressed the infectious properties of the H5N1 flu virus (a decrease in virus titer of 3.0–3.3 log units relative to the control), but did not protect the cell culture against cytopathogenic effects of the influenza A virus at 48 and 72 h after infection. At the same time, the fucoidan showed antiviral activity at an early stage of infection, i.e. during the first 24 h. The application of fucoidan to the cell culture in various concentrations an hour before the virus was introduced resulted in a decrease in the titer of the influenza virus by 2.3–3.3 log units. With simultaneous introduction of influenza A virus and fucoidan into the cell culture, the virus titer was decreased by 2.3–2.8 log units relative to the control. These results open up prospects in terms of developing new approaches to interrupting virus adsorption by sensitive cells.

A number of reports have been devoted to comparative studies of the anti-influenza effectiveness of polysaccharides from several families of algae [81]. Song et al. [82] assessed *Grateloupia filicina* (family Rhodophita), *Ulva pertusa* (family Chlorophyta), and *Sargassum qingdaoense* (Ochrophyta) in their studies. The yield of polysaccharide was 19.7% (*G. filicina*, GFP), 12.1% (*U. pertusa*, UPP), and 7.2% (*S. qingdaoense*, SQP). The content of sulphate groups in the polysaccharide was also different: 13.54% in UPP, 19.89% in GFP, and 5.64% in SQP. The structure of all three polysaccharides was established, and their biological activities were investigated in vivo and in vitro. The safe concentration for SQP and UPP was 5 mg/mL, and for GFP this was 2.5 μg/mL. The in vitro antiviral effects were evaluated against the H9N2 influenza virus. In the hemagglutination test, the most active were UPP and SQP. Under the action of these polysaccharides, the titer of influenza B virus decreased significantly. GFP was the most active in reducing virus replication and SQP was the least active. The most effective dose of the polysaccharide was 20 µg/mL. Using real-time polymerase chain reaction (PCR), it was shown that the expression of the H9N2 gene was significantly reduced under the influence of the studied polysaccharides. The best inhibitory effect was observed with a GFP dose of 20 µg/mL. In the same study, the authors showed that all three polysaccharides had immunomodulatory potential as the studied polysaccharides were active in the spleen lymphocyte proliferation test. The greatest activity in this test was shown by SQP, the effect of which was dose-dependent. Maximum values were obtained when using the sulphated polysacchrides at a dose of 500 µg/mL. In the experimental group of mice treated with the polysaccharide, the levels of IFN-γ and IL-4 were significantly increased (*p* < 0.05). All polysaccharides increased numbers of CD3+ and CD4+ lymphocytes in the blood compared to controls, but only SQP increased the level of CD8+ cells. Thus, the best effect was obtained with the polysaccharide of the brown alga *S. quingdaoense,* especially at a dose of 50 mg/kg. The authors attributed this phenomenon to the presence of fucose residues in its structure, which play a significant role in immunomodulation [83]. The content of fucose in the PCA was 0.02, 0.05, and 1% for UPP, GFP, and SQP, respectively. The authors also observed immunological phenomena as such the proliferation of spleen cells and the humoral immune response, connected with the presence of fucose in these preparations. The more pronounced suppression of the replication of the influenza virus by the GFP polysaccharide was explained by the higher content of sulphate groups in the structure of this polysaccharide [84,85]. The authors suggest the use of all three polysaccharides as a potential alternative to vaccination, as well as to suppress the replication of the influenza virus.

In another study [81], as a result of a comparative study of sulphated polysaccharides activity from algae of different families (red algae: *Polysiphonia lanosa*, *Furcellaria lumbricalis*, and *Palmaria palmate*; brown algae: *Ascophyllum nodosum* and *Fucus vesiculosis*; green alga: *Ulva latuca*), it was found that fucoidans from brown algae, i.e., *F. vesiculosis* and *A. nodosum*, had the highest anti-influenza activity. The total sugar content in the polysaccharides studied varied from 15.4% (*U. latuca*) to 91.4% (*F. lumbricalis*). Galactans (agars or carrageenans) were mainly isolated from *P. lanosa*, xylans from *P. palmate*, and fucoidans from brown algae. Heteropolysaccharides were isolated from green algae.

The interaction between the H5N1 influenza virus and fucoidan were investigated by Bobrovnitsky [86]. For the first time, the authors visualized this process by scanning probe microscopy, which provides information about the surface microrelief and measures the length and height of observed objects. In this case, measurements were made of the height of viral particles before and after treatment with fucoidans from two types of algae. The concentration of fucoidans was 1 and 100 ng/mL. It was found that the average size of the virus particles after treatment with fucoidans changed significantly. In the case of a lower concentration, the average height of the particles increased from 40 to 45 nm. With increasing concentrations, the height of the virus particles reached 50 nm. These results indicate an interaction between the positively charged groups of lipoproteins of the viral envelope and the negatively charged sulphate groups of fucoidans; this interaction may be the cause of the antiviral effect of polysaccharides. The antiviral effect of these compounds is probably due to the encapsulation of viral particles and their deactivation as a result of this. Adhesion of fucoidan on the surface of viral particles is an irreversible process.

Based on this evidence, fucoidans have not only a direct effect on influenza viruses but also affect the processes of viral attachment and replication, interact with neuraminidase, and inhibit the release of viruses from cells [62,80]. In addition, they promote antiviral immunity, enhance antioxidant protection, and reduce the appearance of inflammation. Numerous studies [75,85,87,88] have demonstrated the influence of these compounds on factors important in innate and adaptive immunity, such as the antioxidant system. Another positive quality of fucoidans is their antibacterial action, which in some cases will allow them to be used to prevent bacterial complications, which often aggravates the course of influenza infection.

## 6. Lectins

Lectins are widespread carbohydrate-binding proteins and glycoproteins that can specifically and reversibly non-covalently bind mono- and oligosaccharides, both in solution and localized on the cell surface [89,90]. In this way, lectins contribute the so-called first line of defence against bacteria and viruses. These compounds exhibit high specificity in relation of glycoconjugates of bacteria and viruses. Most studied and characterized lectins have been isolated from higher plants; lectins from algae have been studied less thoroughly. However, the observed antiviral and antitumor effects of these compounds have led scientists to look at them from a new perspective. It was previously known that lectins have two or more carbohydrate-binding sites [91]. Connecting to the surface of microorganisms, the lectin can agglutinate and prevent the spread of pathogens throughout the body. In recent years, new families of lectins have been found in the cyanobacterium *Oscillatoria agardhii* (OAA) [92,93,94,95], as well as in the red algae *Eucheuma serra* and *Kappaphycus alvaresii* (KAA-2) [96]. They usually have two or four tandem repeats consisting of highly conserved sequences, but do not have homology with other protein families. The uniqueness of these lectins is that they bind carbohydrates with exceptionally high specificity for high-mannose (HM) glucans in the trisaccharide core, including Manα(1-3) Manα(1-6)Man. At low nanomolar levels, these lectins have potential antiviral activity against the influenza virus due to the recognition of HM-glucans in the composition of the glycoproteins of the spikes of the influenza virus.

Mu et al. [97] isolated lectin HRL40 from the green alga *Halimeda renschii*, which was highly specific to HM-N-glycans with (1,3)-bound monosaccharide residues. Lectin HRL40, by binding to the hemagglutinin of the virus, effectively inhibited (with an ED_50_ 2.45 nM) the infectious process in NCI-H292 cells caused by the influenza A/H3N2/Udom/72 virus. Additionally, a lectin with anti-influenza activity was obtained by Sato et al. [98] from the red alga *Eucheuma serra*. This compound, called by the authors a “high mannose-specific lectin” and designated as KAA-2, effectively inhibited the entry of the influenza virus into cells. The carbohydrate-binding profile of this lectin was determined by centrifugation and ultrafiltration. KAA-2 was associated exclusively with high mannose N-glycans, but not with other glycans. The authors tested this lectin against various strains of influenza virus, including the pandemic variant H1N1-2009. With the immunofluorescent method, it was shown that lectin prevented the virus from entering the host cells. Using ELISA, it was found that the lectin KAA-2 was directly associated with the HA of the influenza virus. It was proposed the use of this lectin as a future means of preventing influenza infection. 

In the study by Sato et al. [99], the anti-influenza activity of lectins with various carbohydrate specificities was investigated on MDCK cells using different strains (clinical isolates) of the influenza virus (H1N1-2009, A/Oita/ou1P3-3/09). The best results in terms of inhibiting influenza infection were obtained with the HM-binding lectin ESA-2. The EC_50_ in this case was 12.4 nM. This lectin recognized the branched structure of HM-glycans, including the trisaccharide containing Manα(1-3)Manα(1-6)Man in the D2 branch as a primary target. The direct interaction between the lectin ESA-2 with the viral envelope glycoprotein HA was demonstrated by ELISA. This interaction was effectively suppressed by glycoproteins carrying HM-glycans, suggesting that ESA-2 binds to the HA of the influenza virus through HM-glycans. The lectin inhibited the penetration of the virus into cells most effectively when simultaneously introducing the virus and the lectin into the cell culture. When processing ESA-2 cells, viral antigens were not detected in the cells, which indicated that this lectin inhibited the initial stages of virus penetration into the cells. At the same time, no cytopathic effect was observed in infected cells. The antiviral profile of ESA-2 was similar to the lectin KAA-2 from *Kappaphycus alvarezii*, which belongs to the same anti-HIV lectin family [98]. The lectin was non-toxic up to 1000 nM (the highest dose used in this experiment). Sensitivity to lectin depended on the strain of influenza virus. The most susceptible were strains A/Philippines/2/82 (EC_50_ 17.2 ± 3.9) and WSN/33 (EC_50_ 34.6 ± 2.7 nM). In this case it was also proposed to use the lectin ESA-2 in the future as a disinfectant or prophylactic agent.

## 7. Polyphenols of Algae

The composition of the polyphenol fraction of brown algae is characterized by the predominant content of phorotannins, which are unique complex biopolymers of marine origin and the main cytoplasmic components of these hydrobionts. These compounds are contained inside the cell in both the free and bound state [100]. Phlorotannins have antioxidant, hepatoprotective, anti-allergic, anti-tumour, anti-inflammatory, anti-bacterial, and anti-diabetic properties [101]. Ryu et al. [102] purified from the brown alga *Ecklonia cava* phlorotannin, which proved to be an effective selective inhibitor of the NA of influenza virus (72% at a dose of 30 µg/mL). By fractionating the ethyl acetate layer, five phlorotannins were obtained, identified as phloroglucinol, ecol, 7-phloracol, fluorofucofuroecol, and diecol. The inhibitory activities of these components were assessed against influenza viruses from NA group 1 (A/Bervig_Mission/1/18[H1N1]), A/PR/8/34[H1N1]) and group 2 (A/Hong Kong/8/68[H3N2), A/Chicken/Korea/MS96[H9N2]). All five phlorotannin derivatives were found to be selective inhibitors of NA. Fluorofucofuroecol showed the strongest inhibitory activity against NA viruses of group 1 (IC_50_ of 4.5 and 14.7 mmol, respectively); diecol inhibited the NA of influenza virus strains of group 2 more effectively. All derivatives of phlorotannin enhanced the NA-inhibitory effect of ozaltamivir.

## 8. Biopolymers of Algae Are Adjuvants for the Influenza Vaccines

Biopolymers of algae are currently being also investigated as candidate adjuvants for the next generation of influenza vaccines. The main direction of improvement for anti-influenza vaccines is to increase their safety. That is why, from whole virion vaccines, the transition was made to split vaccines, and from there to subunit vaccines. However, immunogenicity is often reduced with highly purified antigens [103]. In this regard, scientists study the polysaccharides of algae, whose influence on the formation of innate and adaptive immunity has been described in numerous papers in Russia and in other countries [75,87,88,104]. In the analysis of adjuvant technologies for the creation of vaccines, preference is given to modifiers of functions of the receptors of innate immunity and their signaling pathways [105]. The sulphate polysaccharides of brown algae have some excellent properties as adjuvants: almost complete absence of toxicity, safety, and excellent biocompatibility [106].

In the mechanisms of action of polysaccharides, which are important for the manifestation of the adjuvant effect, one should highlight the ability to exhibit the properties of TLR agonists of innate immunity cells, designed to recognize microbial pathogen-associated molecules. TLRs are major targets for the development of new adjuvants, and TLR agonists are the most preferred adjuvants for vaccines. In the investigation of the specific interaction of polysaccharides with human TLRs, it was found that fucoidans from algae *Saccharina japonica*, *Saccharina cichorioides*, and *Fucus evanescens* specifically bind TLR2 and TLR4, causing activation of the nuclear factor NF-κB. Subsequent expression of genes of proinflammatory cytokines and interferon-inducible genes promote the activation of immunocompetent cells, and the development of an adaptive immune response to unrelated antigens of the Th1 type [107]. Experimental data demonstrate the adjuvant properties of polysaccharides in relation to various antigens and vaccine strains of infectious agents, including influenza virus [104,108]. The results of our experimental studies also indicate the adjuvant activity of fucoidan from the brown alga *F. evanescens*, manifested as an increase in the immunogenicity of the inactivated influenza virus A/California/7/09 H1N1pdm09. At the same time, the effect of fucoidan was more pronounced compared with the traditional licensed adjuvant aluminum hydroxide. In addition, with repeated immunization of animals, fucoidan provided a reduction in antigenic load. The results indicate the promising application of fucoidan as an adjuvant in vaccines of influenza [109]. Of significant interest are the results of a randomized, double-blind, placebo-controlled study with elderly volunteers, focusing on the ability of fucoidan to have an adjuvant effect when administered orally. The volunteers took the fucoidan from *U. pinnatifida* at a dose of 300 mg/day orally for 4 weeks. Subsequent immunization with the trivalent influenza vaccine led to the identification of higher antibody titers against all strains of the virus contained in the vaccine, compared with antibody titers in individuals who received placebo. In the group of volunteers who received fucoidan, after nine weeks, there was a clear tendency to increase the activity of natural killer cells, and the absence of allergic and other undesirable immune reactions [110]. Despite the positive results of testing of sulphated polysaccharides as adjuvants, it should be taken into account that the use of fucoidans as drugs is currently limited, due to difficulties with obtaining structurally characterized and homogeneous samples or oligomeric fractions of fucoidans. In this regard, active work is underway to obtain low molecular weight polysaccharides or fucooligosaccharides (homo- or hetero-oligosaccharides containing from 2 to 10 monosaccharide residues) related to natural fucoidans. A number of studies have indicated the high immunomodulatory activity of low molecular weight, structurally characterized fractions of fucoidan or its oligosaccharides, but studies on their adjuvant properties are rare [109]. Thus, the adjuvant activity of low molecular weight polysaccharides obtained from the brown alga *F. evanescens* was investigated; this was done using enzymes that provided a stable, reproducible structure. The authors considered that this substance can be used as a pharmaceutical substance or adjuvant as a part of vaccine preparations [109]. Therefore, polysaccharides from brown algae can apparently be used as safe and effective adjuvants in the composition of next generation influenza vaccines. Fucoidans may form a new molecular basis for the creation of immune adjuvants, including for influenza vaccines, due to their high biocompatibility, lack of toxicity, and good tolerance by the human body.

## 9. Conclusions

Last decade, there have been many works devoted to the antiviral potencies of sulphated polysaccharides. The formation of pathogen resistance to drugs on the pharmaceutical market requires new approaches to the treatment of viral diseases, including influenza. To do this, it is necessary to have drugs with different mechanisms of action, which in addition to antiviral effects, have anti-inflammatory, antioxidant, and immunomodulatory activity and to which viruses form resistance only on rare occasions. As presented in this review, the sulphated polysaccharides of brown, red, and green algae have such properties in relation to influenza infection (Figure 2).

The pharmaceutical market currently offers a carrageenan-based spray for local (intranasal) application. There are no other drugs based on polysaccharides, which is associated with difficulties in standardization. To standardize these compounds, the physicochemical parameters such as molecular weight, monosaccharide composition, degree of sulphation and other structural features of polysaccharides should be determined. One approach to solving this problem is to obtain structurally characterized and homogeneous samples of native polysaccharides with a low molecular weight or oligomeric fractions. At the same time, due to the wide spectrum of biological activity of polysaccharides and, most importantly, their ability to exert a virucidal effect, that prevent the penetration of influenza viruses into cells and suppress the replication of viral particles, these unique compounds can be used as the basis for the creation of a new generation of medicines. In addition, the almost complete absence of toxicity and pathogen resistance, relatively low cost, a significant yield of the final product, good solubility, significant reserves of natural sources, and the possibility of cultivation algae make polysaccharides promising candidates for the development of drugs with antiviral activity, in particular anti-influenza activity.

## Figures and Tables

**Figure 1 marinedrugs-17-00373-f001:**
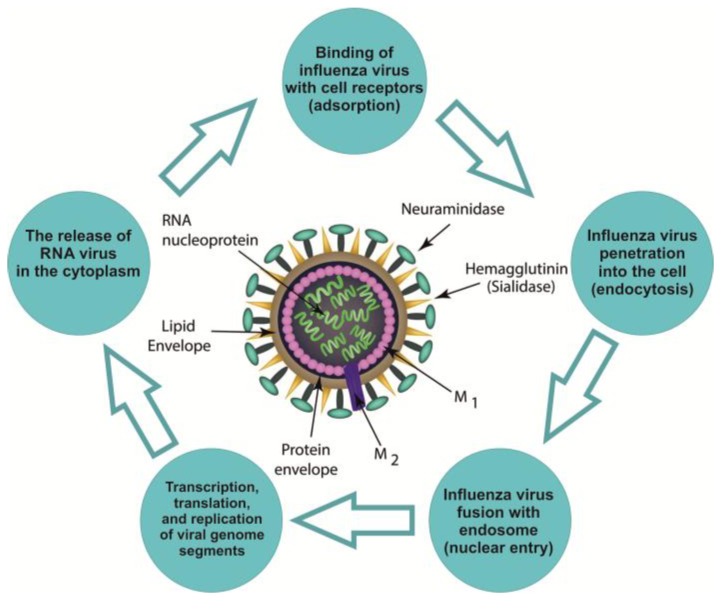
The life cycle of influenza virus.

**Figure 2 marinedrugs-17-00373-f002:**
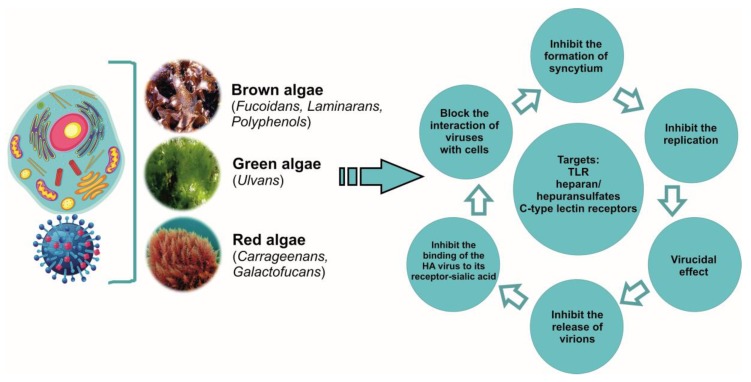
Antiviral activity of algae metabolites.

**Table 1 marinedrugs-17-00373-t001:** Anti-influenza drugs.

Direct Acting Antiviral Drugs	Mechanism of Action of Antiviral Drugs	Side Effects
Oseltamivir (Tamiflu)	Competitive and selective inhibitor of neuraminidase of influenza viruses A and B	The gastrointestinal tract can be involved (nausea and vomiting; diarrhea, abdominal bloating, and fecal incontinence); central nervous system (dizziness, migraine, sleep disturbance, weakness); respiratory tract system (bronchitis, cough, infections of the upper respiratory tract); generalized pain
Zanamivir (Relenza)	Selective inhibitor of neuraminidase of influenza viruses A, inhibitor of replication and release of new virus particles	Allergies, breathing problems, dermatological disorders
Umifenovir (Arbidol)	A specific inhibitor of the fusion of a viral lipid membrane with cell membranes. Interacts with NA, prevents its conformation, which is necessary for the fusion of NA with endosome membranes. Interferon inductor. Stimulator of humoral and cellular immunity	Pruritus, rash, angioedema, urticaria, anaphylaxis
Riamylovir (Triazavirin)	Inhibitor of viral RNA synthesis and replication of genomic fragments	Allergies; the gastrointestinal tract can be involved (nausea and vomiting; diarrhea, abdominal bloating, and fecal incontinence)
Rimantadine, amantadine	M2 channel blockers. An inhibitor of the early stage of virus reproduction from the moment it enters the cell until the beginning of the transcription process. RNA inhibitor	The gastrointestinal tract can be involved (diarrhea); central nervous system (dizziness, migraine, sleep disturbance, weakness); respiratory tract system (cough); generalized pain; allergies

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
