# Peer review of "Metabolites of Seaweeds as Potential Agents for the Prevention and Therapy of Influenza Infection"

_marinedrugs, 2019, doi:10.3390/md17060373_

Round 1
Reviewer 1 Report
The manuscript entitled “Metabolites of seaweeds as potential agents for the prevention and therapy of influenza infection” is well written and described.
In my opinion, the manuscript could be published with minor modifications as below.
1. Abstract is not informative. It is required improvement.
2. Summarize each section with appropriate method such as table or figure. It is more helpful for readers. For example, “section 7. Lectins”, authors described several lectins for potential agent for influenza infections. It could be summarized by table which including characters of lectins.
3. The description of possible mechanisms of metabolites or proteins from seaweed for protection of influenza infection might be helpful to understand an importance of algal extracts. Each antiviral agent involved in different stage of viral disease, replication or infection stages. Sometimes it just related to increase immune system, which means non-directly affect to antiviral mechanisms.
4. Check “line 624” question mark. It may be misspelled.
Author Response
Dear Reviewer, thank you for your carefully reading of our manuscript and critical comments.
Comment. Abstract is not informative. It is required improvement.
Answer. We improve the abstract.
Comment. Summarize each section with appropriate method such as table or figure. It is more helpful for readers. For example, “section 7. Lectins”, authors described several lectins for potential agent for influenza infections. It could be summarized by table which including characters of lectins.
Answer. The review contents information about polysaccharides, lectins, and polyphenols against influenza infection with appropriated references. We did not summarize by table a characteristics of each, because in part of manuscripts authors described only some structural characteristics of substances, also they used substance purified by other author with appropriate references or they used commercial substances.
Comment. The description of possible mechanisms of metabolites or proteins from seaweed for protection of influenza infection might be helpful to understand an importance of algal extracts. Each antiviral agent involved in different stage of viral disease, replication or infection stages. Sometimes it just related to increase immune system, which means non-directly affect to antiviral mechanisms.
Answer. We include the figure with possible mechanisms of metabolites from marine organisms for protection of influenza infection (the figure 2 in conclusion).
Comment.Check “line 624” question mark. It may be misspelled.
Answer. Thank you. We correct it. Was: Manα(1-3)Man(1-6). Corrected: Manα(1-3)Manα(1-6)Man.

Reviewer 2 Report
This review tries to extensively gather data regarding the application of marine metabolites as anti-influenza drugs. Although the pertinence of the subject. it is my strongest believe, that if the information is organised in a different manner, the review paper would be more easily understood, and really represent a tool for researchers that are looking for new line of research or simply data update.
My comments are the following:
1) Introduction: please reorganize this section. It would be easier to understand, if authors started by overall data related with respiratory infections, and then focus on the data related particularly to influenza infection. Please make sure that all claims are supported by references. Also, in this section (or in the following) it would be nice to have the explanation of the methodology used for papers selection for this review (e.g. keywords used in the search, time frame,...)
2) Exciter and section 3: this section is of utmost importance, particularly, if you are also targeting readers that are not experts in the area. I would recommend to provide a simpler and more complete explanation regarding influenza virus composition and life cycle. Please make sure that all explanations are supported by references. Also, make sure that all acronyms are properly explained, the first time it appears. The reason why I advise for a more complete explanation, is related to the fact that in the following sections, its harder to understand where the potential new drugs will act to prevent influenza infection. Indeed, a wonderful add-on to the review could be a figure where the authors could summarise all targets within influenza infection cycle, that could be addressed by marine metabolites.
3) section 4: It could be useful to include a table summarising the available drugs in the market, including target, scope of activity, disadvantages/specificities of each drug. Also, some references are missing after statements.
4) Section 5 onwards: it would be beneficial to have the different marine metabolites chapters organised in similar terms. Also, many of the statements, particularly when generalisations are made, are lacking references. Regarding Figure 1, it would be more fruitful to have a figure where the structures of the polysaccharides that actually have anti-influenza activity would be presented.
Also, regarding carrageenan section, it would be more interesting to have a critical overview and sum up of the most important lessons retrieved from the different papers, rather than presenting the different findings of each study. For instance, summarising the most relevant activity (if it was specific to particular strains or had a more broader activity), routes of administration, possible side effects.
Regarding Table 1, it should have the structures of the particular fucoidans discussed in that particular section (the other structures do not enrich the review paper, representing confusing elements). Also, care was given to explain fucoidan biodistribution in the body. Although this information could be relevant for this polymer clinical application,it seems the paper is losing focus from influenza infection. So this information should be presented along or when it is necessary to assess fucoidan potential as anti-influenza drugs.
The Alginate subchapter is misleading. Authors explain the alginate potential in fighting other virus infection, and then try to suggest that the activity of Orvirem is attributed to the simultaneous application of alginate and rimantadine. This is misleading, as in this case alginate is simply working as an excipient, aiding the drug delivery, but not having any kind of direct influence in influenza infection.
The polyphenols section is unfocused, summarising first the antiviral activity against other virus, and then against influenza. It could be reorganised in a way that clearly emphasises its anti-influenza activity.
The first paragraph of section 9, is summarising some of the information given before, without adding nothing new, and without including refs. Could be simply removed.
The conclusion section is too long, not offering the final readout of the review paper. It could be revised to a stronger impact.
Examples of missing references: lines:31, 35, 62, 65, 68, 72, 77, 88, 104, 118, 138, 160,162, 195,220, 223, 278, 329, 512, 564, 575, 674,
Typos: lines:131, 624, 630
Author Response
Answer to reviewer 2.
Dear Reviewer, thank you for your carefully reading of our manuscript and critical comments.
Comment. Introduction: please reorganize this section. It would be easier to understand, if authors started by overall data related with respiratory infections, and then focus on the data related particularly to influenza infection. Please make sure that all claims are supported by references. Also, in this section (or in the following) it would be nice to have the explanation of the methodology used for papers selection for this review (e.g. keywords used in the search, time frame,...)
Answer.We improved the Introduction, add appropriated references.
1. 1. Lvov, N. I.; Likhopoenko, V. P., Acute respiratory infections. Guide to infectious diseases. StP.:Foliant 2011; 2(III), 7-122.
2. 2. Osidak, L. V.; Drinevskiy, V. P.; Erofeeva, M. K.; Eropkin, M. Y.; Konovalov, N. I.; Smororintsev, E. A.; Golovacheva, E. G.; Donluray, E. A.; Baybus, A. M.; Wojciechowska, E. M.; Tsybalova, L. M., Influenza A (H1N1) 2009 in Russia. Terra Medica nova. Infectious diseases (In Rissian) 2009, 4-5, 6-9.
3. 3. Henrickson, K. J.; Hoover, S.; Kehl, K. S.; Hua, W., National disease burden of respiratory viruses detected in children by polymerase chain reaction. Pediatr Infect Dis J 2004, 23, (1 Suppl), S11-8.
We add information in abstract about materials and methods (lines 29-37).
Comment. Exciter and section 3: this section is of utmost importance, particularly, if you are also targeting readers that are not experts in the area. I would recommend to provide a simpler and more complete explanation regarding influenza virus composition and life cycle. Please make sure that all explanations are supported by references. Also, make sure that all acronyms are properly explained, the first time it appears. The reason why I advise for a more complete explanation, is related to the fact that in the following sections, its harder to understand where the potential new drugs will act to prevent influenza infection. Indeed, a wonderful add-on to the review could be a figure where the authors could summarise all targets within influenza infection cycle, that could be addressed by marine metabolites.
Answer. We add in Review the figures with influenza virus composition and life cycle (figure 1).
Comment. section 4: It could be useful to include a table summarising the available drugs in the market, including target, scope of activity, disadvantages/specificities of each drug. Also, some references are missing after statements.
Answer. We add information about the available drugs in the market (Table 1).
Comment. Section 5 onwards: it would be beneficial to have the different marine metabolites chapters organised in similar terms. Also, many of the statements, particularly when generalisations are made, are lacking references. Regarding Figure 1, it would be more fruitful to have a figure where the structures of the polysaccharides that actually have anti-influenza activity would be presented.
Answer. We summarized and present information about the structural characteristics of polysaccharides that processed antivirus activity in text, because very often authors used commercial polysaccharides without additional information about exactly structure, also they used substance purified by other author with appropriate references or authors published only some structural characteristics (monosaccharides composition, content of sulfates).
Comment Also, regarding carrageenan section, it would be more interesting to have a critical overview and sum up of the most important lessons retrieved from the different papers, rather than presenting the different findings of each study. For instance, summarising the most relevant activity (if it was specific to particular strains or had a more broader activity), routes of administration, possible side effects.
Answer. We add summarized information about carrageenan (lines 407-410 and 458-462).
Comment Regarding Table 1, it should have the structures of the particular fucoidans discussed in that particular section (the other structures do not enrich the review paper, representing confusing elements). Also, care was given to explain fucoidan biodistribution in the body. Although this information could be relevant for this polymer clinical application,it seems the paper is losing focus from influenza infection. So this information should be presented along or when it is necessary to assess fucoidan potential as anti-influenza drugs.
Answer. We removed the table 1 from manuscript.
Comment The Alginate subchapter is misleading. Authors explain the alginate potential in fighting other virus infection, and then try to suggest that the activity of Orvirem is attributed to the simultaneous application of alginate and rimantadine. This is misleading, as in this case alginate is simply working as an excipient, aiding the drug delivery, but not having any kind of direct influence in influenza infection.
Answer. Thank for comment. After discussion we decide to remove information about alginates from the manuscript.
Comment The polyphenols section is unfocused, summarising first the antiviral activity against other virus, and then against influenza. It could be reorganised in a way that clearly emphasises its anti-influenza activity.
Answer. We reorganized the polyphenols section.
Comment The first paragraph of section 9, is summarising some of the information given before, without adding nothing new, and without including refs. Could be simply removed.
Answer. Thank you for comment. We remove the first paragraph of section 9.
Comment The conclusion section is too long, not offering the final readout of the review paper. It could be revised to a stronger impact.
Answer. We correct the conclusion.
Comment Examples of missing references: lines:31, 35, 62, 65, 68, 72, 77, 88, 104, 118, 138, 160,162, 195,220, 223, 278, 329, 512, 564, 575, 674,
Answer. We add appropriated references.
Comment Typos: lines:131, 624, 630
Answer. We correct all typos.

Round 2
Reviewer 2 Report
Dear authors,
although some corrections have been made in the manuscript, some of the issues I early detected are still present in the current manuscript. It is still a hard reading, with unfocused and sometimes repeated information. Still, some references are missing.
Introduction:
lines 62 to 68 are repeated from previous paragraph
"
Sulphated polysaccharides of algae (red, brown, and green), such as carrageenans, fucoidans, and
74 ulvans fulfil these requirements. This review presents the in vitro and in vivo results of experimental
75 and clinical studies demonstrating the excellent effectiveness of these compounds in combating
76 influenza infection and explain the need to use them as a potential basis for the creation of new
77 drugs with a broad spectrum of activity. "
It is misleading stating that sulphated polysaccharides fulfil completely the requirements, and that that possess excellent effectiveness
Exciter: what do you mean by exciter? In this section, the composition and function of each virus component could be reorganised in order to have a full view of a virus particle before entering in how to they multiple...
line 85, what is a RNP cell?
A figure was added, which is enriching the review, but this figure do not properly engage the different virus components in the virus cycle, so it could be improved.
Line 95 a ref is missing
the paragraph that starts in line 110 is confusing, because it starts by explaining why NA is important, but then it moves to HA again. A ref is missing at line 121
Table 1. suggestion: substitute "pains of various locations" by "generalised pain"
Line 208, Glycosaminoglycans have been presented previously, but authors start a phrase by "
Among them there are the glycosaminoglycans, a class"
refs missing in 212 and 221
Carrageenans section: it is too long and can be summarised. The main ideas of each study should be gathered, summarised and discussed as a full, rather than collecting information, and graft them to each other.
It is hard to understand the association between efficacy against dengue virus and influenza virus in paragraph 298
A full stop is missing in 311.
Line 394: typo
Fucoidans section:
refs missing in lines: 464 and 465
The information introduced in line 480 does not explain why derived structures have antiviral properties. Also, the fucoidan pharmacodinamics were introduced in the following paragraph, without adding much to the antiviral properties of fucoidans.
Line 576: CPS activity (is not previously written in full)
Line 618: HM glucans (is not previously written in full)
Conclusion:
authors start conclusion section naming sulphated polysaccharides, however, this review has a broader scope so possibly algae polysaccharides should be named instead.
Author Response
Dear Reviewer, thank you for your carefully reading of our manuscript and critical comments.
Comment. Introduction: lines 62 to 68 are repeated from previous paragraph
Answer. We are sorry for this, it is our mistake. We remove the text lines 62-68.
Comment. Sulphated polysaccharides of algae (red, brown, and green), such as carrageenans, fucoidans, and
74 ulvans fulfil these requirements. This review presents the in vitro and in vivo results of experimental
75 and clinical studies demonstrating the excellent effectiveness of these compounds in combating
76 influenza infection and explain the need to use them as a potential basis for the creation of new
77 drugs with a broad spectrum of activity. "
It is misleading stating that sulphated polysaccharides fulfil completely the requirements, and that that possess excellent effectiveness
Answer. We add the additional information for the explanation (lines 78-82).
Comment. Exciter: what do you mean by exciter? In this section, the composition and function of each virus component could be reorganised in order to have a full view of a virus particle before entering in how to they multiple...
line 85, what is a RNP cell?
Answer. We change the title of this paragraph: Influenza virion. And correct the sentence about RNP (lines 93-95).
Comment. A figure was added, which is enriching the review, but this figure do not properly engage the different virus components in the virus cycle, so it could be improved.
Answer. We add legend for this figure.
Comment. Line 95 a ref is missing
Answer. We add references in line 95.
Comment. the paragraph that starts in line 110 is confusing, because it starts by explaining why NA is important, but then it moves to HA again. A ref is missing at line 121
Answer. Thank you. It is mistyping. We add reference at line 121.
Comment. Table 1. suggestion: substitute "pains of various locations" by "generalised pain"
Answer. Thank you. We substitute "pains of various locations" by "generalised pain"
Comment. Line 208, Glycosaminoglycans have been presented previously, but authors start a phrase by "
Among them there are the glycosaminoglycans, a class"
Answer. We correct it: “Among them there are the glycosaminoglycans, a class" substitute with “The glycosaminoglycans are a class"
Comment. refs missing in 212 and 221
Answer. We add ref. in 212 and 221.
Comment. Carrageenans section: it is too long and can be summarised. The main ideas of each study should be gathered, summarised and discussed as a full, rather than collecting information, and graft them to each other.
Answer. We summarized this section.
Comment. It is hard to understand the association between efficacy against dengue virus and influenza virus in paragraph 298
Answer. “The antiviral effect of ι-carrageenan depends on the type of virus. If, for example, a polysaccharide interferes not only with dengue virus adsorption, but also blocks the fusion process, then in the case of influenza virus infection, the adsorption process is blocked.”
Yes, it means that for different type of virus carragenans act for different processes: only for adsorption (influenza virus infection) or adsorption and the fusion process (dengue virus).
Comment. A full stop is missing in 311.
Answer. Thank you. We correct it.
Comment. Line 394: typo
Answer. Thank you. We correct this word.
Comment. Fucoidans section:
refs missing in lines: 464 and 465
The information introduced in line 480 does not explain why derived structures have antiviral properties. Also, the fucoidan pharmacodinamics were introduced in the following paragraph, without adding much to the antiviral properties of fucoidans.
Answer. We add reference.
Comment. Line 576: CPS activity (is not previously written in full)
Answer. Thank you. We change CPS for full word: sulfated polysaccharide.
Comment. Line 618: HM glucans (is not previously written in full)
Answer. Thank you. We add full word.
Comment. Conclusion:
authors start conclusion section naming sulphated polysaccharides, however, this review has a broader scope so possibly algae polysaccharides should be named instead.
Answer. We change section naming.